# Riemannian Metric Matching for Scalable Geometric Modelling of Distributions

**Jacob Bamberger**
Department of Computer Science
University of Oxford

**Adam Gosztolai**
Institute of Artificial Intelligence
Medical University of Vienna

**Pierre Vandergheynst**
EPFL

**Michael Bronstein**
Department of Computer Science
University of Oxford
AITHYRA

**Iolo Jones**
Department of Computer Science
University of Oxford

## Abstract

High-dimensional datasets often concentrate near low-dimensional structures, but estimating their geometry from samples typically relies on graphs and kernels that scale poorly with dataset size and dimension. We propose Riemannian metric matching: a denoising probabilistic framework for learning the Riemannian geometry of data using neural networks. Specifically, we learn the *carré du champ* operator, which, using diffusion geometry, gives us access to the Riemannian geometry toolkit for downstream machine learning and statistics. Our key observation is that the carré du champ operator can be formulated as a conditional expectation over random perturbations of the data, which can be exploited for sample-wise training and constant cost, amortized inference without explicit kernel construction. To the best of our knowledge, we provide the first neural surrogate that estimates the underlying Riemannian geometry of data with a provable consistency guarantee in the large data limit. Empirically, metric matching rivals or improves the accuracy of $k$-NN-based diffusion geometry estimators, while enabling amortized inference that is up to $400\times$ faster, and supports graph-free geometric analysis on high-dimensional images where nearest neighbors break down.

## 1 Introduction

High-dimensional datasets are typical in machine learning, but empirical evidence suggests that their local structure is effectively low-dimensional, commonly called the manifold hypothesis. This low-dimensional structure has led to the use of geometric or spatial tools on diverse data types, which can probe intrinsic properties of data, like dimensionality, tangent spaces, and curvature. Such methods have been successfully applied in areas ranging from computer vision (Murase & Nayar, 1993) and generative modeling (Song & Ermon, 2019; Bamberger et al., 2026) to single-cell data (Moon et al., 2019) and molecular dynamics (Facco et al., 2017; Diepeveen et al., 2024).

The first step in a standard geometric data analysis pipeline is to build a graph whose nodes correspond to data points and whose edges are often weighted by a kernel function and encode local connectivity. While convergence guarantees hold in the infinite-data limit (Coifman & Lafon, 2006), these approaches become computationally prohibitive at scale. In particular, both computational and memory costs grow super-linearly, if not quadratically, with dataset size. Crucially, these costs persist at inference time, as processing new points require recomputing nearest neighbors for each query, preventing efficient out-of-sample extension. These graph and kernel methods are also limited in their by their reliance on the pairwise distances between points, which are cursed by dimension.

Recent work leverages trained neural networks – such as VAEs or diffusion models – to recover geometric information (Stanczuk et al., 2024), often through the Jacobian of the trained network (Arvanitidis et al., 2018). This avoids pairwise distances and graphs, but shifts the cost to computing Jacobians of large networks, which can be expensive and numerically unstable in high dimensions. Since these networks are typically trained for representation learning or generative modeling rather

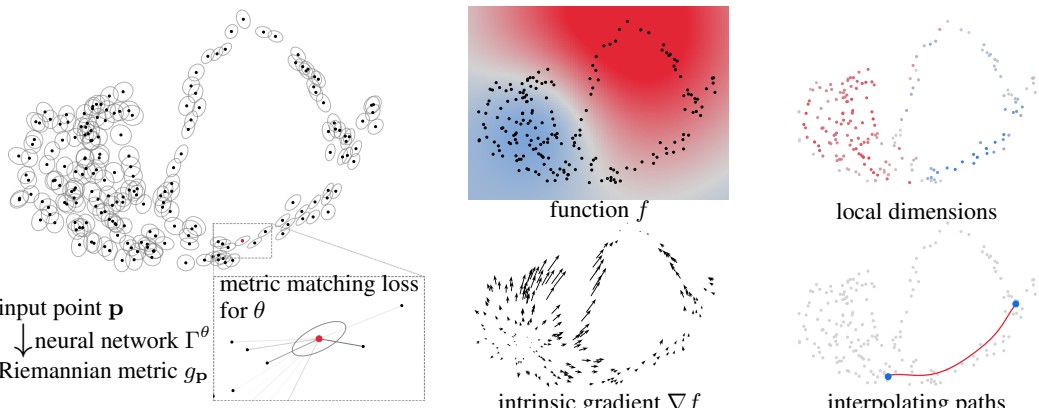

Figure 1: **Intrinsic diffusion geometry with Riemannian metric matching.** The neural network learns a Riemannian metric (ellipses, top left) by implicitly averaging multiple rank-1 contributions through the denoising metric matching loss (inset, bottom left). This gives us access to the toolkit of Riemannian geometry methods via diffusion geometry. Given a scalar field $f$ (top middle), we can compute its intrinsic gradient $\nabla f$ with respect to the data geometry (bottom middle). We estimate local data dimensions from the ratio of the metric eigenvalues (top right: red corresponds to higher dimension), and on-manifold interpolating paths between pairs of points (bottom right).

than geometric estimation, existing approaches provide few theoretical guarantees that the recovered geometric quantities converge to the true underlying geometry in the infinite-data limit.

In this work, we combine the strengths of both approaches: retaining the theoretical guarantees of kernel methods while leveraging deep neural networks for scalable training and inference. Our main contribution is a new training objective inspired by the denoising objectives from diffusion models (Sohl-Dickstein et al., 2015; Ho et al., 2020; Song et al., 2021). This loss allows a neural network to learn a Riemannian metric at every point by rewriting an intractable quantity as the marginalization of tractable ones. Specifically, we learn an estimate for the *carré du champ* operator (Bakry et al., 2013), which can be used to recover the Riemannian geometry of the data via diffusion geometry (Jones, 2024a;b; Jones & Lanners, 2026). Given finite data, the minimizer of the loss corresponds to a classical kernel-based estimate of the carré du champ, and in the infinite-data limit, recovers the desired quantity when the data distribution is locally supported on a manifold. This allows the design of novel deep learning methods by directly inheriting tools from Riemannian geometry.

**Diffusion geometry and the carré du champ.**    Classical Riemannian geometry applies on manifolds, but real-world data rarely satisfy the stringent conditions of being a manifold (like having no branching points, and a uniformly constant local dimension). Diffusion geometry (Jones, 2024a) describes the geometry of more general spaces through the behavior of diffusion processes on them. This applies to the low-dimensional but non-manifold data geometries that occur in practice, allowing the transfer of Riemannian geometry methods to generic data geometries.

On a Riemannian manifold $(\mathcal{M}, g)$, the canonical diffusion is Brownian motion $dx = dB_t$, whose evolution is characterized by its *infinitesimal generator*, the Laplace-Beltrami operator $\Delta_g$, which is related to the Riemannian metric $g$ by the carré du champ (CDC) identity

$$g(\nabla f, \nabla h) = \frac{1}{2}\left(f\Delta_g h + h\Delta_g f - \Delta_g(fh)\right) \tag{1}$$

where $f, h : \mathcal{M} \to \mathbb{R}$ are two smooth scalar fields. In the theory of Markov diffusion operators (Bakry et al., 2013), the infinitesimal generator $\mathcal{L}$ of any Markov process defines a bilinear form called the *carré du champ operator*

$$\Gamma_\mathcal{L}(f, h) := \frac{1}{2}(f\mathcal{L}h + h\mathcal{L}f - \mathcal{L}(fh)) \tag{2}$$

so that $\Gamma_{\Delta_g}(f, h) = g(\nabla f, \nabla h)$ on a manifold. In diffusion geometry, this relationship is abstracted and used to define a generalized Riemannian geometry relative to an arbitrary Markov diffusion process. By learning the carré du champ, we can access the toolkit of classical Riemannian geometry methods, such as intrinsic calculus, curvature, and topology (Jones, 2024a;b; Jones & Lanners, 2026).

## 2    CONDITIONAL METRIC MATCHING

We aim to learn the geometry of an underlying dataset by observing samples $X \sim p$, and estimating the carré du champ operator. When the data lie on a manifold, a standard approach to approximate $\Delta_g$, uses the heat kernel $w_\varepsilon(\mathbf{x}, \mathbf{y}) = \exp(-\|\mathbf{x} - \mathbf{y}\|^2/2\varepsilon)$ with bandwidth $\varepsilon$. This approach was originally introduced to justify kernel methods (Coifman & Lafon, 2006) and will also serve as the basis to metric matching. The heat kernel can be used to define the normalized diffusion operator

$$(P_\varepsilon f)(\mathbf{y}) := \frac{(K_\varepsilon f)(\mathbf{y})}{d_\varepsilon(\mathbf{y})},$$

where $K_\varepsilon f)(\mathbf{y}) := \mathbb{E}_{X \sim p(x)}[w_\varepsilon(\mathbf{y}, X)f(X)]$ and $d_\varepsilon(\mathbf{y}) := \mathbb{E}_{X \sim p(x)}[w_\varepsilon(\mathbf{y}, X)]$ are the kernel operator and a corresponding degree function, respectively. Intuitively, $(P_\varepsilon f)(\mathbf{y})$ is a local average of $f$ near $\mathbf{y}$ under a Gaussian window of scale $\varepsilon$. When $p$ is supported on a manifold $\mathcal{M}$, the operator $\mathcal{L}_\epsilon := (\mathbf{I} - P_\varepsilon)/\varepsilon$ converges pointwise to $c\mathcal{L}$ as $\varepsilon \to 0$, where $c$ is a constant, and $\mathcal{L}f := \Delta_g f - 2g(\nabla \log p, \nabla f)$, consisting of the second-order diffusion term $\Delta_g$ and a first-order drift term depending on the score of the distribution (Coifman & Lafon, 2006). We can substitute $\mathcal{L}_\epsilon = (\mathbf{I} - P_\varepsilon)/\varepsilon$ into Eq. 2 to rewrite

$$\Gamma_\varepsilon(f, h)(\mathbf{y}) = \frac{\mathbb{E}_X\left[w_\varepsilon(\mathbf{y}, X)\big(f(X) - f(\mathbf{y})\big)\big(h(X) - h(\mathbf{y})\big)\right]}{2\varepsilon\, \mathbb{E}_X[w_\varepsilon(\mathbf{y}, X)]}, \tag{3}$$

which can be interpreted as the local covariance between $f$ and $h$ under the scale-$\varepsilon$ diffusion generated by $P_\varepsilon$. These expectations can be estimated from samples using kernel-weighted neighborhood graphs, but this becomes computationally and memory intensive for many samples, and suffers from the curse of dimensionality in high dimensions. In the next section, we show how to construct a scalable loss function and training procedure that allows a neural network to estimate $\Gamma_\varepsilon(f, h)$ from samples.

**Conditional carré du champ matching.**    Our goal is to train a neural network to compute or approximate $\Gamma_\varepsilon$. A natural objective is to regress $\Gamma_\varepsilon$ under the smoothed density $p_Y = p * \mathcal{N}(0, \varepsilon\mathbf{I})$:

$$\mathcal{L}_{marg}^{CDC}(\theta) := \mathbb{E}_Y\left[\left(\Gamma_\varepsilon^\theta(Y) - \Gamma_\varepsilon(f, h)(Y)\right)^2\right]. \tag{4}$$

However, as we have seen, $\Gamma_\varepsilon(f, h)(\mathbf{y})$ from Eq. 3, which we will refer to as the *marginal* CDC, is intractable. We therefore introduce a tractable *conditional* CDC, defined as

$$\Gamma_\varepsilon(f, h)(\mathbf{x}, \mathbf{y}) := \frac{1}{2\varepsilon}\left(f(\mathbf{x}) - f(\mathbf{y})\right)\left(h(\mathbf{x}) - h(\mathbf{y})\right).$$

Since we chose $p_Y = p * \mathcal{N}(0, \varepsilon\mathbf{I})$, we have $p_Y(\mathbf{y}|\mathbf{x}) = (2\pi\varepsilon)^{-D/2}w_\varepsilon(\mathbf{y}, \mathbf{x})$ and $d_\varepsilon(\mathbf{y}) = (2\pi\varepsilon)^{D/2}\mathbb{E}_X[p_Y(\mathbf{y}|X)] = (2\pi\varepsilon)^{D/2}p_Y(\mathbf{y})$ so we can rewrite the marginal CDC as

$$\Gamma_\varepsilon(f, h)(\mathbf{y}) = \frac{\mathbb{E}_X\left[p_Y(\mathbf{y}|X)\Gamma_\varepsilon(f, h)(X, \mathbf{y})\right]}{p_Y(\mathbf{y})} = \mathbb{E}_X\left[\Gamma_\varepsilon(f, h)(X, Y) \mid Y = \mathbf{y}\right]. \tag{5}$$

We now introduce the tractable conditional loss:

$$\mathcal{L}_{cond}^{CDC}(\theta) := \mathbb{E}_{X, Y|X}\left[\left(\Gamma_\varepsilon^\theta(Y) - \Gamma_\varepsilon(f, h)(X, Y)\right)^2\right] \tag{6}$$

where $X \sim p$ and $Y|X \sim \mathcal{N}(X, \varepsilon\mathbf{I})$. This loss can be sampled from, evaluated, and backpropagated through easily since $X$ is sampled from the data distribution $p$, $Y$ is a noisy version of $X$, and the conditional CDC is computed in $\mathcal{O}(1)$. The expectation decomposes over independent samples $X$ and noise $Y|X$, so is trivially parallelizable and well suited for modern GPU-based mini-batch and distributed training. Perhaps surprisingly, we prove that the conditional loss is equal to the marginal loss up to a constant independent of the parameters $\theta$, so the gradients of both losses are equal.

**Theorem 2.1**  $\mathcal{L}_{cond}^{CDC}(\theta) = \mathcal{L}_{marg}^{CDC}(\theta) + C$, $C$ *is independent of* $\theta$, *so* $\nabla_\theta \mathcal{L}_{cond}^{CDC}(\theta) = \nabla_\theta \mathcal{L}_{marg}^{CDC}(\theta)$.

**Conditional Riemannian metric matching.**    The loss in Eq. 6 can be applied to any scalar fields $f$ and $h$. However, in practice we will train a network to regress the CDC $\Gamma_\varepsilon(x_i, x_j)$ of each pair of coordinate functions $x_k : \mathcal{M} \to \mathbb{R}$, and we show in Sec. D.1 that recovers the CDC $\Gamma_\varepsilon(f, h)$ of

any pair $f, h : \mathcal{M} \to \mathbb{R}$ by the chain rule. Applying the conditional CDC loss to all the coordinate functions $f = x_i$ and $h = x_j$ yields matrix-valued targets and predictions, giving the Frobenius loss

$$\mathcal{L}_{cond}^{\text{Riem}} := \mathop{\mathbb{E}}_{X, Y|X} \left[ \left\| \Gamma_\varepsilon^\theta(Y) - \frac{1}{2\varepsilon}(X - Y)(X - Y)^T \right\|_F^2 \right], \tag{7}$$

where $\Gamma_\varepsilon^\theta$ outputs a positive semi-definite $D \times D$ matrix approximating the CDC matrix. Since $\varepsilon$ is a hyperparameter, we can condition the network on $\varepsilon$, allowing to capture the geometry of the data across all scales simultaneously. The full training algorithm can be found in Algorithm. 2.

**Convergence results.** While the losses in Eq. 6 and Eq. 7 are very simple and scalable, they also come with theoretical guarantees in the case that that data is locally sampled from a manifold. We present extended theoretical guarantees in Appendix D, and here show the practically relevant result that when the loss in Eq. 7 is minimized, and $p$ is locally supported on a manifold, then the output of the neural network approximates the projection onto the tangent space as $\varepsilon \to 0$.

**Corollary 2.2** *Let $\mu$ be a probability measure on $\mathbb{R}^D$ and $x \in supp(\mu)$. Suppose that $B(x, \delta) \cap supp(\mu)$ is a manifold (of any dimension, possibly depending on $x$), for some $\delta > 0$. Then the matrix $(\mathbf{\Gamma}_\varepsilon(\mathbf{p}))_{k\ell}$ converges to the projection matrix onto the tangent space $\mathbb{R}^D \to T_{\mathbf{p}}\mathcal{M}$.*

## 3 EXPERIMENTS

We apply metric matching on synthetic and real-world datasets to evaluate i) its scalability to large datasets, ii) its accuracy in controlled settings with ground truth, and iii) its scalability to high dimensions. First, we use synthetic datasets with a ground-truth geometry to assess the accuracy of intrinsic dimension and tangent space estimation, as well as the scalability of the method with sample size. Second, we apply our approach to high-dimensional image data to demonstrate training and evaluation at scales where classical graph-based geometric methods become impractical. We provide additional experimental details in Appendix G.

**Synthetic setting: scalability and accuracy.** We begin by sampling $N$ points uniformly from the $d$-dimensional sphere embedded in $\mathbb{R}^D$. We vary $N$ to evaluate the scalability of metric matching relative to baselines. We use the ground-truth tangent spaces to evaluate tangent space prediction. We set $d = 8$, $D = 64$, and a fixed architecture across all dataset sizes: an MLP with 15M parameters predicting a rank $r = 16$ matrix via low-rank training.

**Scalability.** We first evaluate the scalability of the neural surrogate for CDC-based geometric estimators against $k$-NN-based CDC estimators, and report throughput (number of inferences per second) as a function of the number of samples. We compare both the raw CDC computation and eigen-decomposition to identify the leading $d$ eigenvectors, as is typical in tangent space prediction. The raw neural CDC surrogate is advantageous over the raw CDC estimator at approximately 16k points, and is over $82\times$ faster for 2M points and $400\times$ faster for 8M points. In the small-data regime, the computational bottleneck for tangent space prediction is eigendecom-position: low-rank training substantially reduces this cost by replacing a $D \times D$ eigendecomposition with an $r \times r$ one, where $r \ll D$, leading to speedups of $67\times$, $132\times$, $194\times$, and $359\times$ for 8k, 2M, 4M, and 8M points, respectively.

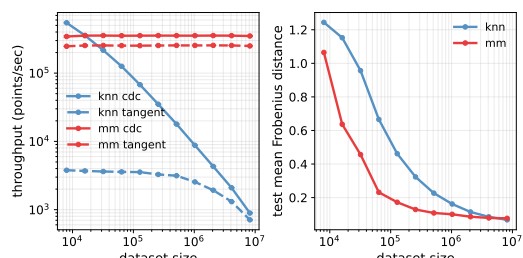

Figure 2: Comparison of trained metric matching (mm) network against $k$-NN-based baseline, on throughput (left) and performance (right). Experiments were run on a NVIDIA A10 (24GB) GPU.

**Accuracy.** We evaluate the accuracy of tangent space prediction by measuring the Frobenius distance $\|UU^T - \hat{U}\hat{U}^T\|_F$ on a test set, where $U \in \mathbb{R}^{d \times D}$ denotes a basis for the ground-truth tangent space and $\hat{U}$ is the estimator obtained by eigendecomposition. The neural surrogate clearly outperforms $k$-NN for datasets smaller than 4M, with $46\%$ improved performance relative to the

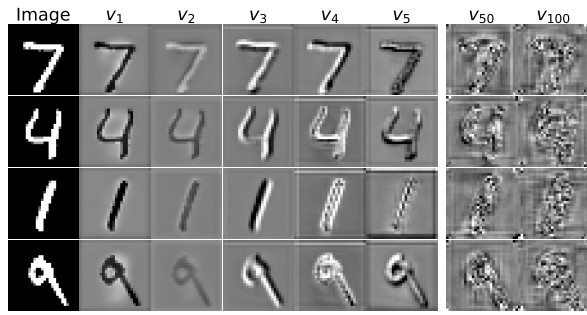 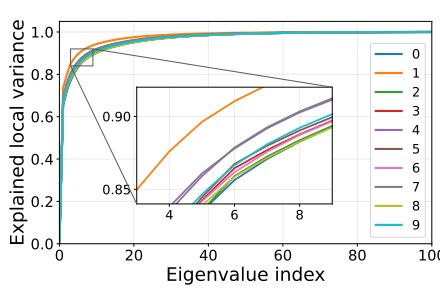

(a) Visualization of the first 5, 50th, and 100th eigenvectors of the CDC matrix learned by metric matching.

(b) Class-wise mean cumulative eigenvalues.

Figure 3: MNIST CDC eigen-analysis.

$k$-NN baseline, while closely matching it for larger datasets. We visualize the eigenvalues in Fig. 4, where the neural surrogate accurately captures 8 leading dimensions, while the $k$-NN CDC captures 9. These results indicate better generalization of the neural surrogate than the $k$-NN–based CDC.

**High-dimensional images.** Kernel methods tend to suffer from the curse of dimensionality, restricting their scalability to high-dimensional datasets. We test metric matching in this setting by attempting to recover the intrinsic geometry of the $784$ dimensional image dataset MNIST, consisting of 60k training and 10k validation images, which are black and white images representing digits $0$ to $9$. We train a standard UNet architecture with the low-rank metric matching objective Eq. 8 with rank 100. We visualize the quality of the learned geometry by plotting the eigenvectors of the CDC at validation images in Fig. 3a. We see that the first eigenvectors are highly interpretable, compared to the 50th and 100th which appear much noisier.

In particular, the second eigenvector is close to a damped version of the original image, while the third and fourth are concentrated at edges and so generate translations. We evaluate the importance of each tangent direction by its eigenvalue, and we plot the cumulative eigenvalue distribution (i.e. local explained variance) for each class in Fig. 3b. We observe a sharp dropoff in eigenvalues, reflecting the low-dimensional structure. The digit 1 has a faster drop, suggesting a lower intrinsic dimension, which is perhaps due to its relative visual simplicity.

## 4 CONCLUSION

We presented metric matching: a tractable denoising-style objective for estimating the carré du champ with neural networks. This provides a scalable alternative to graph CDC estimators, enabling out-of-sample geometry prediction in a single forward pass. On synthetic manifolds, metric matching improves the accuracy and throughput compared to classical $k$-NN baselines. On high-dimensional image data, metric matching remains effective where $k$-NN methods fail due to the curse of dimensionality, thereby expanding the practical reach of diffusion geometry tools to high-dimensional data. We hypothesize that this robustness stems from the inductive biases of the neural estimator: characterizing when and why such generalization occurs is an important direction for future work. We hope that this work can provide a platform for developing novel deep learning methods that directly inherit tools from Riemannian geometry.

## 5 ACKNOWLEDGEMENTS

This work is partially supported by the EPSRC Turing AI World-Leading Research Fellowship No. EP/X040062/1 and EPSRC AI Hub No. EP/Y028872/1.

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

## A    RELATED WORK

**Geometry in diffusion and generative models.**    Recent work has highlighted deep connections between diffusion models and differential geometry. Stanczuk et al. (2024) show that, at small noise levels $\sigma$, the score $\nabla \log p_\sigma$ aligns with directions normal to the data manifold, and exploit this for intrinsic dimension estimation. Kadkhodaie et al. (2024) study the generalization properties of denoisers through the eigendecomposition of the Jacobian of the score model (the Hessian $\nabla^2 \log p_\sigma$ of the log-density), revealing a rapid spectral decay and semantically meaningful leading eigenvectors. More recently, Kharitenko et al. (2025) formalize this connection by showing that $\mathbf{I} - \sigma^{-2}\nabla^2 \log p_\sigma$ converges to the tangent space projector, which can be accessed via automatic differentiation of the score, and apply this to Riemannian optimization on data manifolds. Our work complements and extends this line of research by explicitly linking modern denoising objectives to classical diffusion maps (Coifman & Lafon, 2006). Our denoising-style loss directly regresses the carré du champ, providing the data-dependent Riemannian metric through a single forward pass while also guaranteeing convergence to the tangent-space projector.

**(Riemannian) metric learning.**    Metric learning originally aimed to learn task-adapted distances for $k$-NN classification and clustering, extending beyond the standard Euclidean metric (Friedman, 1994; Xing et al., 2002; Bellet et al., 2015). More recently, there has been a surge of interest in a Riemannian formulation, which would additionally permit intrinsic geometric notions such as geodesics and curvature. Most such methods either attempt to learn the metric of the embedded data

manifold or to design a metric that is meaningful for a downstream task. Metric learning methods can be broadly grouped into unsupervised, supervised, and implicit approaches. Unsupervised methods infer geometry from samples via graphs or kernels methods, e.g. (Arvanitidis et al., 2016; Jones, 2024b). Supervised methods use a parametric model trained using a loss function based on labels, such as contrastive (Huang et al., 2014) or triplet losses (Weinberger & Saul, 2009). Implicit methods define geometry as a by-product of a learned model, e.g. via pullback metrics induced by the Jacobian of a VAE or embedding model (Arvanitidis et al., 2018; Diepeveen et al., 2025). This Riemannian approach has recently been applied to generative modeling (Kapusniak et al., 2024; Bamberger et al., 2026) and data analysis (Diepeveen et al., 2024), extending beyond its original use in clustering (Xing et al., 2002) and classification (Friedman, 1994). We refer to Gruffaz & Sassen (2025) for a comprehensive survey. Our work builds on this literature by introducing a self-supervised objective for learning the intrinsic geometry of data manifolds, with theoretical guarantees at optimality.

## B    POSITIVE SEMI-DEFINITE AND LOW RANK TRAINING

Since $\Gamma_\varepsilon(Y)$ is symmetric and positive semidefinite (PSD) by definition, we first parameterize a neural network to produce a $D \times D$ matrix $M_\varepsilon^\theta(Y)$ and make it symmetric PSD by setting $\Gamma_\varepsilon^\theta(Y) = M_\varepsilon^\theta(Y)^T M_\varepsilon^\theta(Y)$. Since $D$ can be much larger than the intrinsic dimension, we also consider a low rank version where $M_\varepsilon^\theta(Y) \in \mathbb{R}^{r \times D}$ making $\Gamma_\varepsilon(Y)$ symmetric PSD with a rank upper bounded by the hyperparameter $r$. In this case we observe that the Frobenius norm in Eq. 7 can be simplified by expanding

$$\left\| M_\varepsilon^\theta(Y)^T M_\varepsilon^\theta(Y) - \frac{1}{2\varepsilon}(X - Y)(X - Y)^T \right\|_F^2$$
$$= \left\| M_\varepsilon^\theta(Y)^T M_\varepsilon^\theta(Y) \right\|_F^2 + \frac{1}{4\varepsilon^2} \left\| X - Y \right\|^4$$
$$- \frac{1}{\varepsilon} \left\| M_\varepsilon^\theta(Y)(X - Y) \right\|^2.$$

Since $\left\| M_\varepsilon^\theta(Y)^T M_\varepsilon^\theta(Y) \right\|_F^2 = \left\| M_\varepsilon^\theta(Y) M_\varepsilon^\theta(Y)^T \right\|_F^2$, we can compute this loss without ever materializing any $D \times D$ matrices, considerably improving the memory and time complexity. The second term also does not depend on $\theta$, so we form the simplified low-rank loss

$$\mathcal{L}_{LR} = \mathop{\mathbb{E}}_{X,Y|X} \left[ \left\| M_\varepsilon^\theta(Y) M_\varepsilon^\theta(Y)^T \right\|_F^2 \right]$$
$$- \frac{1}{\varepsilon} \mathop{\mathbb{E}}_{X,Y|X} \left[ \left\| M_\varepsilon^\theta(Y)(X - Y) \right\|^2 \right] \tag{8}$$

If the manifold $\mathcal{M}$ has dimension $d$, we need to take $r$ to be at least $r$, but cannot generally assume that $r = d$ is enough to fully factorize the metric (this would require, at least, the condition that $\mathcal{M}$ is *parallelizable*[1]). However, we prove in Sec. D.2 that picking $r = 2d - 1$ is always enough for the low-rank factorization to be valid.

If a strict upper bound of $r$ to the intrinsic dimensionality is not desired, we consider the efficient parameterization $\Gamma_\varepsilon^\theta(Y) = M_\varepsilon^\theta(Y)^T M_\varepsilon^\theta(Y) + \lambda \mathbf{I}$ where $\lambda$ is a small Tikhonov regularization term ensuring strict positive definiteness. In this case, we get $\mathcal{L}_{LR}^\lambda = \mathcal{L}_{LR} + 2\lambda \| M_\varepsilon^\theta(Y) \|_F^2$, see Appendix C for a derivation.

---

[1]For example, the 2-dimensional sphere embedded in 3 dimensions is not parallelizable due to the hairy ball theorem, so $r = 2$ is not sufficient in that case.

## C    LOW RANK LOSS DERIVATION

When the metric is parameterized as low rank with small identity perturbation $\Gamma_\varepsilon^\theta = M_\varepsilon^{\theta T} M_\varepsilon^\theta + \lambda \mathbf{I}$, and let $\Delta = Y - X \sim \mathcal{N}(0, \varepsilon \mathbf{I})$ be the noise, then:

$$\left\| M_\varepsilon^{\theta T} M_\varepsilon^\theta + \lambda \mathbf{I} - \frac{1}{2\varepsilon} \Delta \Delta^T \right\|_F^2 = \mathrm{Tr}\left( \left( M_\varepsilon^{\theta T} M_\varepsilon^\theta + \lambda \mathbf{I} - \frac{1}{2\varepsilon} \Delta \Delta^T \right)^2 \right) \tag{9}$$

$$= \mathrm{Tr}\left( \left( M_\varepsilon^{\theta T} M_\varepsilon^\theta + \lambda \mathbf{I} \right)^2 + \left( \frac{1}{2\varepsilon} \Delta \Delta^T \right)^2 - \frac{1}{\varepsilon} \left( M_\varepsilon^{\theta T} M_\varepsilon^\theta + \lambda \mathbf{I} \right) \Delta \Delta^T \right) \tag{10}$$

$$= \mathrm{Tr}\left( \left( M_\varepsilon^{\theta T} M_\varepsilon^\theta \right)^2 \right) + 2\lambda \, \mathrm{Tr}\left( M_\varepsilon^{\theta T} M_\varepsilon^\theta \right) + \lambda^2 D + \mathrm{Tr}\left( \left( \frac{1}{2\varepsilon} \Delta \Delta^T \right)^2 \right) \tag{11}$$

$$- \frac{1}{\varepsilon} \left( \mathrm{Tr}\left( M_\varepsilon^{\theta T} M_\varepsilon^\theta \Delta \Delta^T \right) + \lambda \, \mathrm{Tr}\left( \Delta \Delta^T \right) \right). \tag{12}$$

Where we expanded the squares. Now we notice that $\mathrm{Tr}\left( (\Delta \Delta^T)^2 \right) = \mathrm{Tr}\left( \Delta \Delta^T \Delta \Delta^T \right) = \|\Delta\|_2^2 \, \mathrm{Tr}\left( \Delta \Delta^T \right) = \|\Delta\|_2^4$, and due to the cyclic property of the trace (i.e. $\mathrm{Tr}(ABC) = \mathrm{Tr}(CAB)$), we have $\mathrm{Tr}\left( M_\varepsilon^{\theta T} M_\varepsilon^\theta \Delta \Delta^T \right) = \mathrm{Tr}\left( \Delta^T M_\varepsilon^{\theta T} M_\varepsilon^\theta \Delta \right) = \mathrm{Tr}\left( \left( M_\varepsilon^\theta \Delta \right)^T M_\varepsilon^\theta \Delta \right) = \|M_\varepsilon^\theta \Delta\|_2^2$. Reorganizing the terms we get:

$$\left\| M_\varepsilon^{\theta T} M_\varepsilon^\theta + \lambda \mathbf{I} - \frac{1}{2\varepsilon} \Delta \Delta^T \right\|_F^2 = \mathrm{Tr}\left( \left( M_\varepsilon^{\theta T} M_\varepsilon^\theta \right)^2 \right) + 2\lambda \, \mathrm{Tr}\left( M_\varepsilon^{\theta T} M_\varepsilon^\theta \right) - \frac{1}{\varepsilon} \|M_\varepsilon^\theta \Delta\|_2^2 \tag{13}$$

$$+ \underbrace{\lambda^2 D + \frac{1}{4\varepsilon^2} \|\Delta\|_2^4 - \frac{\lambda}{\varepsilon} \|\Delta\|_2^2}_{\text{independent of } \theta} \tag{14}$$

$$= \|M_\varepsilon^{\theta T} M_\varepsilon^\theta\|_F^2 + 2\lambda \|M_\varepsilon^\theta\|_F^2 - \frac{1}{\varepsilon} \|M_\varepsilon^\theta \Delta\|_2^2 + C. \tag{15}$$

Hence we get the low rank with Tikhonov regularization loss $\mathcal{L}_{LR}^\lambda = \|M_\varepsilon^{\theta T} M_\varepsilon^\theta\|_F^2 + 2\lambda \|M_\varepsilon^\theta\|_F^2 - \frac{1}{\varepsilon} \|M_\varepsilon^\theta \Delta\|_2^2$ and setting $\lambda = 0$ yields the low rank loss $\mathcal{L}_{LR} = \|M_\varepsilon^{\theta T} M_\varepsilon^\theta\|_F^2 - \frac{1}{\varepsilon} \|M_\varepsilon^\theta \Delta\|_2^2$. Note that since $\|M_\varepsilon^{\theta T} M_\varepsilon^\theta\|_F^2 = \mathrm{Tr}\left( M_\varepsilon^{\theta T} M_\varepsilon^\theta M_\varepsilon^{\theta T} M_\varepsilon^\theta \right) = \mathrm{Tr}\left( \left( M_\varepsilon^\theta M_\varepsilon^{\theta T} \right) \left( M_\varepsilon^\theta M_\varepsilon^{\theta T} \right) \right)$, and $\|M_\varepsilon^\theta\|_F^2 = \mathrm{Tr}\left( M_\varepsilon^\theta M_\varepsilon^{\theta T} \right)$, both terms can be computed without ever materializing any $D \times D$ matrices.

## D    CONVERGENCE AND THE RIEMANNIAN TOOLKIT

For all $\varepsilon$, the CDC $\Gamma_\varepsilon$ defines a data-driven Riemannian metric which can be used to compute geometric objects via diffusion geometry. In this section, we provide a theoretical guarantee for our construction. Namely, when the data is locally sampled from a manifold and the loss is minimized, then $\Gamma_\varepsilon$ converges to the true metric as $\varepsilon \to 0$. To the best of our knowledge, this is the first such theoretical guarantee for a neural Riemannian metric.

### D.1    RECOVERING THE METRIC FROM AN EMBEDDED MANIFOLD

Our main theoretical guarantee is provided by the following theorem. We note that the manifold assumption only needs to hold locally around a point $x \in \mathcal{M}$, so the result extends to non-manifold data such as disjoint unions of manifolds or settings where the intrinsic dimension may vary across regions of the space. All proofs can be found in Appendix E.

**Theorem D.1 (Convergence of the carré du champ)** *Let $\mu$ be a probability measure on $\mathbb{R}^D$ and $x \in supp(\mu)$. Suppose that $B(x, \delta) \cap supp(\mu)$ is a manifold (of any dimension, possibly depending*

*on x), for some $\delta > 0$, with induced Riemannian metric $g$, and that $\mu$ has a smooth density on this manifold. Then $\Gamma_\varepsilon(f, h)(x) \to g_x(\nabla f, \nabla h)$ as $\varepsilon \to 0$, for all smooth functions $f, h$.*

**Theorem D.2 (Convergence of the carré du champ)** *Let $\mu$ be a probability measure on $\mathbb{R}^D$ and $x \in supp(\mu)$. Suppose that $B(x, \delta) \cap supp(\mu)$ is a manifold (of any dimension, possibly depending on x), for some $\delta > 0$, with induced Riemannian metric $g$, and that $\mu$ has a smooth density on this manifold. Then $\Gamma_\varepsilon(f, h)(x) \to g_x(\nabla f, \nabla h)$ as $\varepsilon \to 0$, for all smooth functions $f, h$.*

**Carré du champ of the ambient coordinates converges to tangent space projection.** As a special case, we consider the ambient coordinate functions $x_k : \mathcal{M} \to \mathbb{R}$, $x_k(\mathbf{x}) = (\mathbf{x})_k$, for $k = 1, \ldots, D$. Applying the carré du champ to these functions yields, at each point $\mathbf{p} \in \mathcal{M}$, the matrix

$$(\mathbf{\Gamma}_\varepsilon(\mathbf{p}))_{k\ell} \approx g_{\mathbf{p}}(\nabla x_k, \nabla x_\ell),$$

where the approximation becomes exact when $\epsilon \to 0$. Let

$$G(\mathbf{p}) = (\Gamma(x_k, x_l)(\mathbf{p}))_{k\ell}$$

be the $D \times D$ carré du champ matrix of the ambient coordinates at $\mathbf{p}$, which corresponds to the pullback of the ambient Euclidean metric to the tangent space. Because $\mathcal{M}$ is embedded in $\mathbb{R}^D$ *isometrically*, the eigenvalues of $G(\mathbf{p})$ are either 1 or 0, with eigenvectors pointing in the tangent and normal directions, respectively, and so $G(\mathbf{p})$ the projection matrix $\mathbb{R}^D \to T_{\mathbf{p}}\mathcal{M}$. We immediately obtain the following corollary by applying Theorem D.2 to the entries of the matrix $\mathbf{\Gamma}_\varepsilon(\mathbf{p})_{k\ell}$.

**Corollary D.3** *Under the same assumptions as Theorem D.2, the matrix $(\mathbf{\Gamma}_\varepsilon(\mathbf{p}))_{k\ell}$ converges (in every matrix norm) to the projection matrix onto the tangent space $\mathbb{R}^D \to T_{\mathbf{p}}\mathcal{M}$.*

**Carré du champ of the ambient coordinates is all you need.** The matrix $\mathbf{\Gamma}_\varepsilon(\mathbf{p})$ of the CDC of the ambient coordinates is important because it suffices to evaluate the CDC of arbitrary functions. If $f, h : \mathbb{R}^D \to \mathbb{R}$ are smooth, then

$$g_{\mathbf{p}}(\nabla f, \nabla h) = \sum_{k,l=1}^{D} \frac{\partial f}{\partial x_j}(\mathbf{p}) \frac{\partial h}{\partial x_l}(\mathbf{p}) g_{\mathbf{p}}(\nabla x_k, \nabla x_\ell)$$

by the chain rule, so

$$\Gamma(f, h)(\mathbf{p}) = \sum_{k,l=1}^{D} \frac{\partial f}{\partial x_k}(\mathbf{p}) \frac{\partial h}{\partial x_l}(\mathbf{p}) \Gamma(x_k, x_\ell)(\mathbf{p}).$$

If we denote the Jacobians of $f, h$ by $\partial f(\mathbf{p})$ and $\partial h(\mathbf{p})$, as in **??**, then this is just the matrix conjugation

$$\Gamma(f, h)(\mathbf{p}) = \partial f(\mathbf{p})^T \mathbf{\Gamma}_\varepsilon(\mathbf{p}) \partial h(\mathbf{p}). \tag{16}$$

We can formalize this in the following corollary.

**Corollary D.4** *Under the same assumptions as Theorem D.2, we have $\partial f(\mathbf{p})^T \mathbf{\Gamma}_\varepsilon(\mathbf{p}) \partial h(\mathbf{p}) \to g_{\mathbf{p}}(\nabla f, \nabla h)$ as $\varepsilon \to 0$, for all smooth $f, h$.*

### D.2 LOW-RANK TRAINING IS VALID FOR $r \geq 2d - 1$

As discussed in Appendix B, the low-rank factorization of the CDC can have topological obstructions if $r$ is too small. We now prove that, for at least $r \geq 2d - 1$, the minimizer of the low-rank loss will still converge to the correct metric $G(\mathbf{p})$.

**Theorem D.5 (Low-rank training is valid for $r \geq 2d - 1$)** *If $r \geq 2d - 1$ and $M_\varepsilon^\theta$ minimises the low-rank loss (Eq. 8) then $M_\varepsilon^\theta(\mathbf{p})^T M_\varepsilon^\theta(\mathbf{p}) \to G(\mathbf{p})$ as $\varepsilon \to 0$.*

### D.3 INTRINSIC GRADIENTS

The carré du champ gives us direct access to the *intrinsic* gradients $\nabla f$ of functions $f$ defined on the data, with respect to the underlying geometry. If $\mathcal{M} \subseteq \mathbb{R}^D$ is an (isometrically embedded) submanifold , we can represent $\nabla f$ in ambient coordinates by its *pushforward* from $\mathcal{M}$ to $\mathbb{R}^D$. At a each point $\mathbf{p} \in \mathcal{M}$, this is given by the vector

$$(g_{\mathbf{p}}(\nabla f, \nabla x_1), ..., g_{\mathbf{p}}(\nabla f, \nabla x_D)) \in \mathbb{R}^D \tag{17}$$

where $g_{\mathbf{p}}(\nabla f, \nabla x_i)$ is the directional derivative of the function $x_i$ along $\nabla f$ at the point $\mathbf{p} \in \mathcal{M}$. By Eq. 1, these terms are just the CDC $g_{\mathbf{p}}(\nabla f, \nabla x_i) = \Gamma(f, x_i)$. When $f : \mathbb{R}^D \to \mathbb{R}$ is defined on the ambient space (such as a neural network or an ambient potential), we can apply Eq. 16 and the fact that $\boldsymbol{\partial} x_i(\mathbf{p}) = \mathbf{e}_i$, to write

$$\begin{aligned}\Gamma(f, x_i)(\mathbf{p}) &= \boldsymbol{\partial} x_i(\mathbf{p})^T \mathbf{\Gamma}_\varepsilon(\mathbf{p}) \boldsymbol{\partial} f(\mathbf{p}) \\ &= (\mathbf{\Gamma}_\varepsilon(\mathbf{p}) \boldsymbol{\partial} f(\mathbf{p}))_i \end{aligned} \tag{18}$$

for each $i = 1, ..., D$. Substituting Eq. 18 into Eq. 17, we can compute the intrinsic gradient $\nabla f$ by the simple matrix-vector product $\mathbf{\Gamma}_\varepsilon(\mathbf{p}) \boldsymbol{\partial} f(\mathbf{p}) \in \mathbb{R}^D$. We apply this method in the middle column of Fig. 1.

## E PROOFS

**Theorem 2.1** $\mathcal{L}_{cond}^{CDC}(\theta) = \mathcal{L}_{marg}^{CDC}(\theta) + C$, $C$ *is independent of* $\theta$, *so* $\nabla_\theta \mathcal{L}_{cond}^{CDC}(\theta) = \nabla_\theta \mathcal{L}_{marg}^{CDC}(\theta)$.

Let $Z := \Gamma_\varepsilon(f, h)(X, Y)$ be the random variable defined by $X \sim p$ and $Y \mid X \sim \mathcal{N}(X, \varepsilon I)$, so Eq. 5 says that $\mathbb{E}[Z \mid Y] = \Gamma_\varepsilon(f, h)(Y)$. We can apply the law of total expectation and bias-variance decomposition to derive

$$\begin{aligned}\mathcal{L}_{cond}^{CDC}(\theta) &= \mathbb{E}\left[(\Gamma_\varepsilon^\theta(Y) - Z)^2\right] \\ &= \mathbb{E}\left[\mathbb{E}\left[(\Gamma_\varepsilon^\theta(Y) - Z)^2 \mid Y\right]\right] \\ &= \mathbb{E}\left[\left(\Gamma_\varepsilon^\theta(Y) - \mathbb{E}[Z \mid Y]\right)^2 + \mathrm{Var}(Z \mid Y)\right] \\ &= \mathbb{E}\left[\left(\Gamma_\varepsilon^\theta(Y) - \Gamma_\varepsilon(f, h)(Y)\right)^2\right] + \mathbb{E}\left[\mathrm{Var}(Z \mid Y)\right] \\ &= \mathcal{L}_{marg}^{CDC}(\theta) + \mathbb{E}\left[\mathrm{Var}(Z \mid Y)\right],\end{aligned}$$

so the result holds with $C = \mathbb{E}\left[\mathrm{Var}(Z \mid Y)\right]$.

**Theorem D.2 (Convergence of the carré du champ)** *Let* $\mu$ *be a probability measure on* $\mathbb{R}^D$ *and* $x \in supp(\mu)$. *Suppose that* $B(x, \delta) \cap supp(\mu)$ *is a manifold (of any dimension, possibly depending on* $x$), *for some* $\delta > 0$, *with induced Riemannian metric* $g$, *and that* $\mu$ *has a smooth density on this manifold. Then* $\Gamma_\varepsilon(f, h)(x) \to g_x(\nabla f, \nabla h)$ *as* $\varepsilon \to 0$, *for all smooth functions* $f, h$.

We would first like to show that, by possibly reducing $\delta$ to any $0 < \delta' < \delta$, we can assume that $\mathcal{M} = B(x, \delta) \cap supp(\mu)$ is a compact manifold with boundary. By assumption, $\mathcal{M}$ is a smooth manifold of $\mathbb{R}^D$, so the ambient distance function $y \mapsto \|y - x\|$ restricts to a smooth function $d_x : \mathcal{M} \to \mathbb{R}$. In particular, $\mathcal{M} = d_x^{-1}[0, \delta)$, and $\mathcal{M}' = d_x^{-1}[0, \delta']$ is a compact submanifold of $\mathbb{R}^D$ with boundary (that is also a submanifold of $\mathcal{M}$), and satisfies all the other conditions on $\mathcal{M}$. We can now apply the *diffusion maps* theorem (Coifman & Lafon, 2006; Belkin & Niyogi, 2003) to the operator $\mathcal{L}_\epsilon := (\mathbf{I} - P_\varepsilon)/\varepsilon$ on $\mathcal{M}$. Let us write $g$ for the induced Riemannian metric on $\mathcal{M}$, $\Delta_g$ for its Laplace-Beltrami operator, and $p$ for the smooth density of $\mu$ with respect to the Riemannian volume form of $(\mathcal{M}, g)$. Then, for smooth $f$, Theorem 2 in (Coifman & Lafon, 2006) (applied here with $\alpha = 0$) states that $|\mathcal{L}_\epsilon f - \mathcal{L} f| \to 0$ as $\varepsilon \to 0$, where

$$\mathcal{L} f = \frac{\Delta_g(fp) - \Delta_g(p)f}{p}.$$

We can apply the carré du champ identity Eq. 1 to write

$$\Delta_g(fp) - \Delta_g(p)f = \Delta_g(f)p - 2g(\nabla p, \nabla f),$$

so

$$\mathcal{L}f = \Delta_g f - 2\frac{g(\nabla p, \nabla f)}{p} = \Delta_g f - 2g(\nabla \log p, \nabla f).$$

We can now again use the carré du champ identity to expand

$$|\Gamma_\varepsilon(f,g) - g_x(\nabla f, \nabla h)| = \frac{1}{2}\big|f(\mathcal{L}_\varepsilon h - \Delta_g h) + h(\mathcal{L}_\varepsilon f - \Delta_g f) - (\mathcal{L}_\varepsilon(fh) - \Delta_g(fh))\big| \quad (19)$$

and notice that each term satisfies

$$\mathcal{L}_\varepsilon h - \Delta_g h = \mathcal{L}_\varepsilon h - \mathcal{L}h + \mathcal{L}h - \Delta_g h = (\mathcal{L}_\varepsilon h - \mathcal{L}h) - 2g(\nabla \log p, \nabla h),$$

so Eq. 19 becomes

$$\frac{1}{2}\big|f[\mathcal{L}_\varepsilon h - \mathcal{L}h] + h[\mathcal{L}_\varepsilon f - \mathcal{L}f] - [\mathcal{L}_\varepsilon(fh) - \mathcal{L}(fh)] - 2fg(\nabla \log p, \nabla h) - 2hg(\nabla \log p, \nabla f) + 2g(\nabla \log p, \nabla(fh))\big|.$$

We can apply the Leibniz rule to

$$-2fg(\nabla \log p, \nabla h) - 2hg(\nabla \log p, \nabla f) + 2g(\nabla \log p, \nabla(fh)) = 2g(\nabla \log p, \nabla(fh) - f\nabla h - h\nabla f) = 0,$$

and so find that

$$|\Gamma_\varepsilon(f,g) - g_x(\nabla f, \nabla h)| \leq \frac{1}{2}\big(|f||\mathcal{L}_\varepsilon h - \mathcal{L}h| + |h||\mathcal{L}_\varepsilon f - \mathcal{L}f| + |\mathcal{L}_\varepsilon(fh) - \mathcal{L}(fh)|\big) \to 0$$

as $\varepsilon \to 0$, since $|\mathcal{L}_\epsilon f - \mathcal{L}f| \to 0$ and $f, h$ are bounded (which is implied by the fact that $\mathcal{M}$ is compact).

## F    TRAINING ALGORITHMS

---

**Algorithm 1** Minibatch training with per-sample noise for Conditional CDC Matching (scalar)

---

**Require:** Dataset $\mathcal{D} = \{x_i\}_{i=1}^N \subset \mathbb{R}^D$; batch size $B$; noise-scale sampler $p(\varepsilon)$; fixed $f, h : \mathbb{R}^D \to \mathbb{R}$; network $\Gamma^\theta : \mathbb{R}^D \times \mathbb{R}_+ \to \mathbb{R}$; optimizer Opt.

1: **for** each training step **do**
2:     Sample minibatch $X = \{X_b\}_{b=1}^B \sim \mathcal{D}$.
3:     For each $b$: sample $\varepsilon_b \sim p(\varepsilon)$.
4:     For each $b$: sample $Y_b \sim \mathcal{N}(X_b, \varepsilon_b \mathbf{I})$.
5:     Compute $T_b \leftarrow \big(f(X_b) - f(Y_b)\big)\big(h(X_b) - h(Y_b)\big)/\varepsilon_b$         // conditional CDC target
6:     Predict $P_b \leftarrow \Gamma^\theta(Y_b, t_b)$
7:     Loss $\mathcal{L} \leftarrow \frac{1}{B}\sum_{b=1}^B (P_b - T_b)^2$
8:     Update $\theta \leftarrow \mathrm{Opt}(\theta, \nabla_\theta \mathcal{L})$
9: **end for**

---

**Algorithm 2** Minibatch training with per-sample noise for Conditional Riemannian Metric Matching (matrix)

---

**Require:** Dataset $\mathcal{D} = \{x_i\}_{i=1}^N \subset \mathbb{R}^D$; batch size $B$; noise-scale sampler $p(\varepsilon)$; network $\Gamma^\theta : \mathbb{R}^D \times \mathbb{R}_+ \to \mathbb{S}^D$ (predicts symmetric $D \times D$); optimizer Opt.

1: **for** each training step **do**
2:     Sample minibatch $X = \{X_b\}_{b=1}^B \sim \mathcal{D}$.
3:     For each $b$: sample $\varepsilon_b \sim p(\varepsilon)$ .
4:     For each $b$: sample $Y_b \sim \mathcal{N}(X_b, \varepsilon_b \mathbf{I})$.
5:     Set $\Delta_b \leftarrow X_b - Y_b$.
6:     Target $T_b \leftarrow (\Delta_b \Delta_b^\top)/\varepsilon_b \in \mathbb{R}^{D \times D}$         // conditional metric target
7:     Predict $P_b \leftarrow \Gamma^\theta(Y_b, \varepsilon_b) \in \mathbb{R}^{D \times D}$
8:     Loss $\mathcal{L} \leftarrow \frac{1}{B}\sum_{b=1}^B \|P_b - T_b\|_F^2$
9:     Update $\theta \leftarrow \mathrm{Opt}(\theta, \nabla_\theta \mathcal{L})$
10: **end for**

---

## G EXPERIMENTAL DETAILS

**Compute.** All experiments were done on a single NVIDIA A10 (24GB) GPU.

### G.1 BANDWIDTH SAMPLING

In practice we exprimented with two different noise/bandwidth $\varepsilon$ sampling strategies, namely the uniform noise sampling, and log-normal distribution, following the noise scheduling strategy of Karras et al. (2022). Specifically, for the log-normal strategy we draw $\log \varepsilon \sim \mathcal{N}(P_{\mathrm{mean}}, P_{\mathrm{std}}^2)$ with $P_{\mathrm{mean}} = -1.2$ and $P_{\mathrm{std}} = 1.2$, and clamp the resulting values to the interval $[\varepsilon_{\min}, \varepsilon_{\max}]$. This choice biases sampling toward smaller bandwidths, which are critical for resolving fine-scale geometric structure.

### G.2 NOISE LEVEL ENCODING

To condition the network on the bandwidth/noise scale $\varepsilon$, we use Fourier feature embeddings of $t$ of the form $e_\omega(\varepsilon) = (\cos(\varepsilon\omega), \sin(\varepsilon\omega))$. We use $d/2$ frequencies given by $\omega_k = T_{\max}^{-\frac{k}{d/2}}$ for $k = 0, \ldots, \frac{d}{2} - 1$, following standard practice in diffusion and flow-matching models.

### G.3 SYNTHETIC SPHERES EXPERIMENT

**FiLM-conditioned residual MLP encoder.** For the synthetic sphere experiments, we parameterize the low-rank factor $M_\varepsilon^\theta(\mathbf{y}) \in \mathbb{R}^{r \times D}$ with a FiLM-conditioned residual MLP (Perez et al., 2018), denoted `MLPEncoder`. The network takes as input a point $x \in \mathbb{R}^D$ and a per-sample noise/bandwidth scalar $\varepsilon$, and outputs

$$\texttt{MLPEncoder}(h, x) \in \mathbb{R}^{r \times D}.$$

This output is used in the low-rank CDC parameterization $\Gamma_\varepsilon^\theta(\mathbf{y}) = M_\varepsilon^\theta(\mathbf{y})^\top M_\varepsilon^\theta(\mathbf{y})$ (or equivalently in the low-rank loss that avoids materializing $\Gamma_\varepsilon^\theta$).

**Output bias.** Optionally, we add an output bias $b_{\mathrm{out}} \in \mathbb{R}^{r \times D}$ to $M_\varepsilon^\theta(\mathbf{y})$ before making $\Gamma_\varepsilon^\theta$, which we find help during early training.

**Noise-scale conditioning.** We condition the network on $\varepsilon$ using a sinusoidal noise-level embedding as describe in Sec. G.2 followed by a small MLP. Concretely, we compute feed the noise-level encoding in a 2-layer MLP with SiLU nonlinearity. This embedding is injected into *every* residual block via FiLM (feature-wise linear modulation) (Perez et al., 2018), rather than concatenated only once at the input.

**Initialization.** We initialize each residual block near the identity map, meaning a zero initialization of the projection heads of each blocks. Importantly, we do *not* zero-initialize the final projection head: $W_{\mathrm{out}}$ is initialized from a zero-mean Gaussian with small standard deviation, and $b_{\mathrm{out}} = 0$. This avoids an initial output $M_\varepsilon^\theta(\mathbf{y}) \approx 0$, which can lead to weak gradients when optimizing losses involving $M^\top M$.

**Synthetic sphere configuration.** Unless otherwise specified, we use ambient dimension $D = 64$ and sphere dimension $d = 8$. The architecture used hidden width $H = 1024$, $L = 4$ residual blocks, rank $r = 16$, and time/noise embedding enabled. We train with AdamW (learning rate $10^{-4}$, no weight decay) for 3000 epochs with batch size 1024 and gradient clipping with norm 1. In this configuration the MLP has roughly 15M parameters. The noise level sampling strategy used during training was log-normal with $h_{\min} = 10^{-4}$ and $h_{\max} = 16$.

**Evaluation and hyper-parameter tuning for Fig. 2.** We select evaluation hyper-parameters via grid search on a held-out validation set, independently for each training-set size. For metric matching, the only tuned hyper-parameter is the conditioning bandwidth $\varepsilon$; we evaluate $\varepsilon = 2^{-k}$ for $k = -9, -8, \ldots, 4$ and report test performance using the value that minimizes the validation error. For the $k$-NN graph baseline, we tune both the number of neighbors and the bandwidth, searching

over $k = 2^p$ $p = 3, 4 \ldots, 11$ and $\varepsilon = 2^{-k}$ for $k = -9, -8, \ldots, 4$, and again report test performance for the configuration achieving the lowest validation error.

## G.4 MNIST EXPERIMENTS

**UNet backbone.** We use a standard timestep-conditioned UNet backbone with residual blocks and attention from (Dhariwal & Nichol, 2021). After the final UNet layer, the network outputs a tensor of shape $[B, K, W, H]$, where $B$ is the batch size, $W$ and $H$ are spatial dimensions, and $K$ is the number of output channels. We then reshape this output to $[B, K, WH]$ (or equivalently flatten the spatial dimensions) and interpret the channel dimension as a collection of $K$ per-pixel feature maps. In our metric-matching parameterization, these $K$ channels represent the entries of a low-rank factor: we set $K = Cr$ where $C$ is the input channels (so ambient dimension $D = CWH$) and reshape the flattened output into a matrix $U_\theta(x) \in \mathbb{R}^{D \times r}$ (per input), where $D$ is the ambient dimension of the data and $r$ is the chosen rank. We then form the predicted (co)metric as the low-rank PSD matrix

$$\hat{\Gamma}_\theta(x) \;=\; U_\theta(x)\, U_\theta(x)^\top \;\in\; \mathbb{R}^{D \times D}.$$

Finally, we optionally add a learned output bias at the UNet head: after the last convolution and before reshaping into $U_\theta(x)$, we add a trainable vector $b \in \mathbb{R}^K$ to the $K$ output channels and broadcast it across spatial dimensions, followed by the reshaping described above.

**MNIST configuration.** For MNIST we use $28 \times 28$ inputs with `model_channels=` 64, `num_res_blocks=` 2, `channel_mult`$(1, 2, 2)$, and enable attention at downsampling factor 4 (i.e., $7 \times 7$ feature maps). We additionally include a learnable output bias term which we initialize a centered normal distribution with $1e - 3$ variance, as it empirically improves optimization for early stages of training. We use an output rank of $r = 100$ for the low rank parameterization, and use low rank training. We use Adam optimizer with learning rate $2 \times 10^{-4}$, $\beta_1 = 0.9$, $\beta_2 = 0.999$, and $\epsilon = 10^{-8}$, and no weight decay. Training is run for up to 1500 epochs with batch size 128. We apply gradient clipping with global $\ell_2$-norm threshold 1.0. Each training step samples a noise/scale parameter $h \in [h_{\min}, h_{\max}]$ with $h_{\min} = 10^{-4}$ and $h_{\max} = 25$ using a uniform sampling scheme. We trained using gaussian augmentation with 0.1 standard deviation.

**Inference parameters.** We use a bandwidth of $\varepsilon = 7$ as conditioning bandwidth in order to generate Fig. 3a, Fig. 3b and **??**. For the Riemannian optimization experiment we applied natural gradient descent with constant learning rate 0.01 and $2^{13}$ steps. Additionally, we add a slight isotropic component to the metric replacing $\Gamma^\theta$ by $(1 - \delta)\Gamma^\theta + \delta\mathbf{I}$ with $\delta = 10^{-4}$ in order to allow for small movement in all direction, as it improved convergence of the algorithm.

## H ADDITIONAL RESULTS

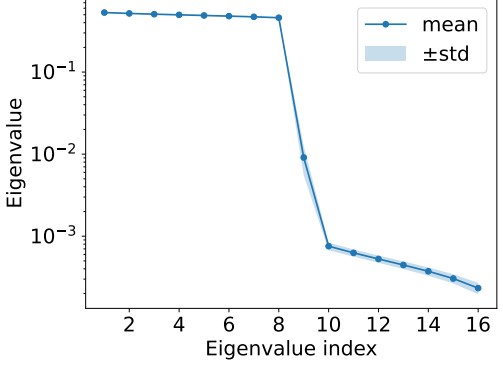 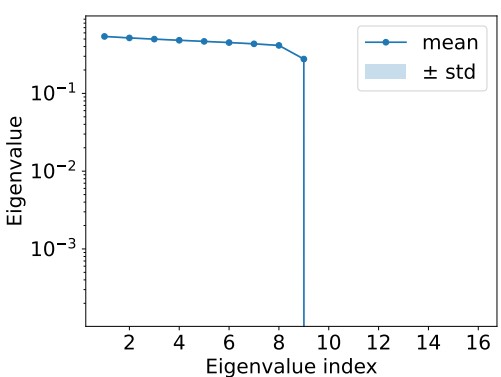

(a) Metric matching MLP (best bandwidth $2^{-5}$).

(b) $k$-NN CDC estimator (best bandwidth $2^{-2}$ and $k = 1024$).

Figure 4: Mean ordered top $16$ eigenvalues on the $8$-dimensional sphere (512k samples), for the best-performing tangent space predictor hyperparameters of (left) metric matching and (right) a $k$NN-based CDC estimator.

