# OpenReview forum: "Riemannian Metric Matching for Scalable Geometric Modelling of Distributions"
_ICLR.cc/2026/Workshop/GRaM — ICLR 2026 Workshop GRaM Poster_

### Official Review · Reviewer_ZcB8 · 2026-02-23
**A neat method for metric matching inspired by classical diffusion maps.**

**Rating:** 9
**Confidence:** 4

**Review:**

**Strengths:**

The paper is very well-written, despite the amount of material in it. A very nice work. I also appreciate the authors' motivation to approach the very difficult problem of manifold estimation from the perspective of diffusion maps, given the recent advances in using diffusion frameworks within modern machine learning.

I particularly enjoyed Theorem 2.1, which is an encouraging result that connects diffusion maps to more modern score-matching and diffusion approaches. Experimentally, the authors evaluate their proposed model in interesting settings, showing it achieves what they set out to do. While I missed an experiment showing the use of this framework to learn a lower-dimensional embedding of the data, I think this goes to show how interesting the work itself is.

**Weaknesses:**

In the experiments, the synthetic setting seems to be using a model which is quite large for the setting. In my opinion, this makes it unclear whether the results are due to the model capacity or the quality of the approach. Specifically in terms of scalability, I think the results are quite unsurprising and could be left for the appendix. More specifically, I would suggest moving Figure 4 up to take that spot and move the scalability results down.

**Minor comments:**

- l. 44: "in their by their"
- l. 95: missing left (
- l. 145: simplea nd -> simple and
- l. 471-472: Missing ref.

**Pmlr Suitability:**

NA

---

### Official Review · Reviewer_D1py · 2026-02-24

**Rating:** 8
**Confidence:** 4

**Review:**

The authors propose a technique to learn a Riemannian metric for data drawn from a distribution in the ambient space that is expected to exhibit manifold-like structure. In particular, the density is not assumed to be everywhere concentrated near a low-dimensional manifold, rather, the intrinsic dimensionality of the support may vary across different regions. The authors rely on the carré du champ identity and the graph Laplacian to construct an objective function, which is then used to train a specifically designed neural network to learn a Riemannian metric approximating the structure of the data. The paper provides theoretical results supporting the proposed objective, as well as experimental results demonstrating the effectiveness of the approach.

The topic of the paper is highly relevant to the direction of the workshop. It focuses on the timely problem of modeling the geometry of an underlying data manifold from finite observations, which typically do not lie exactly on a low-dimensional manifold but are instead sampled from a probability density in the ambient space. The work builds on the renewed interest in diffusion geometry

The technical aspects of the paper appear sound, but I have not checked all the proofs in detail, and the modeling approach is relatively straightforward. The experimental section supports the proposed theory reasonably well. Nevertheless, additional experiments across diverse settings would strengthen the empirical support for the method.

The paper is in general well-written, but I think that some parts can be be further explanated (see questions below).

Questions
1. The derivation and notation of Eq. 5 are somewhat unclear. I assume the idea is that, instead of using all training data X for each test point y, it is sufficient to use only the training points within the kernel support. Is this correct?

2. Eq. 3 (and later the more efficient Eq. 7) suggests that, in practice, the learned Riemannian metric approximates the local covariance matrix of the training data. In the limit $n\to \infty$ and $\varepsilon\to 0$, should the Riemannian metric on the manifold in the ambient space correspond to a covariance matrix with d eigenvalues equal to 1 and D-d eigenvalues equal to 0? Here it is not as an artifact of the finite data and the non-vanishing kernel bandwidth?

3. The discussion of the coordinate functions f and h before Eq. 7 is somewhat unclear. Since y is not assumed to lie exactly on a manifold, but rather in the ambient space around it, it would be helpful to elaborate further on the role of these coordinate functions and how they contribute to recovering the metric. I suppose that the goal is simply to select the components of the learned covariance. The discussion in the appendix (Sec. D) is not sufficiently informative in this regard. Including illustrative figures in two or three dimensions (similar to Fig. 4 left) could clarify the different quantities involved. In particular, since the Laplace–Beltrami operator is intrinsic to the manifold, the use of ambient-space coordinate functions complicates the setting and some additional explanation would be beneficial.

4. While the theoretical motivation and the resulting objective (modeling the Riemannian metric as a covariance matrix) are convincing, the metric is trained only in regions close to the training data. What is the behavior of the learned metric away from these regions? How does it generalize in areas with little or no data support?

**Pmlr Suitability:**

NA

---

### Meta-Review · Area_Chair_L3Ce · 2026-02-24

**Decision:**

Accept

**Metareview:**

The reviewers are very positive about the paper saying it's well-written with nice theory and good experiments.

**Relevance To Proceedings:**

Tiny paper — does not apply

**Relevance To Workshop:**

Yes — suitable for GRaM

---

### Decision · Program_Chairs · 2026-03-02

Accept (Poster)